# Subclinical thyroid dysfunction and depressive symptoms: protocol for a systematic review and individual participant data meta-analysis of prospective cohort studies

Lea Wildisen,[1] Elisavet Moutzouri,[1,2] Shanthi Beglinger,[1,2] Lamprini Syrogiannouli,[1] Anne R Cappola,[3] Bjørn O Åsvold,[4,5] Stephan J L Bakker,[6] Graziano Ceresini,[7] Robin Dullaart,[6] Luigi Ferrucci,[8] Hans Grabe,[9] J Wouter Jukema,[10] Matthias Nauck,[11,12] Stella Trompet,[13] Henry Völzke,[9] Rudi G J Westendorp,[14] Jacobijn Gussekloo,[13,15] Robin P Peeters,[16] Stefan Klöppel,[17] Drahomir Aujesky,[2] Douglas C Bauer,[18] Nicolas Rodondi,[1,2] Cinzia Del Giovane,[1] Martin Feller,[1,2] for the Thyroid Studies Collaboration

For numbered affiliations see end of article.

**Correspondence to**
Lea Wildisen;
lea.wildisen@biham.unibe.ch

## ABSTRACT

**Introduction** Prospective cohort studies on the association between subclinical thyroid dysfunction and depressive symptoms have yielded conflicting findings, possibly because of differences in age, sex, thyroid-stimulating hormone cut-off levels or degree of baseline depressive symptoms. Analysis of individual participant data (IPD) may help clarify this association.

**Methods and analysis** We will conduct a systematic review and IPD meta-analysis of prospective studies on the association between subclinical thyroid dysfunction and depressive symptoms. We will identify studies through a systematic search of the literature in the Ovid Medline, Ovid Embase, Cochrane Central Register of Controlled Trials (CENTRAL) and Cumulative Index to Nursing and Allied Health Literature (CINAHL) databases from inception to April 2019 and from the Thyroid Studies Collaboration. We will ask corresponding authors of studies that meet our inclusion criteria to collaborate by providing IPD. Our primary outcome will be depressive symptoms at the first available individual follow-up, measured on a validated scale. We will convert all the scores to the Beck Depression Inventory scale. For each cohort, we will estimate the mean difference of depressive symptoms between participants with subclinical hypothyroidism or hyperthyroidism and control adjusted for depressive symptoms at baseline. Furthermore, we will adjust our multivariable linear regression analyses for age, sex, education and income. We will pool the effect estimates of all studies in a random-effects meta-analysis. Heterogeneity will be assessed by I². Our secondary outcomes will be depressive symptoms at a specific follow-up time, at the last available individual follow-up and incidence of depression at the first, last and at a specific follow-up time. For the binary outcome of incident depression, we will use a logistic regression model.

**Ethics and dissemination** Formal ethical approval is not required as primary data will not be collected. Our findings

## Strengths and limitations of this study

► This will be the first systematic review and meta-analysis with individual participant data (IPD) to assess the association between subclinical thyroid dysfunction and depressive symptoms.
► By using IPD instead of aggregate data, we can define thyroid dysfunction based on the same criteria across all included studies, which will reduce between-study heterogeneity, and we can analyse subgroups without ecological fallacy.
► We will investigate the association between subclinical thyroid dysfunction and depressive symptoms in different subgroups (age, sex, prior depressive symptoms and different levels of thyroid-stimulating hormone).
► A possible limitation of this study could be that studies used different scales to measure depressive symptoms, so we will convert all the scores to the Beck Depression Inventory scale to make the scale consistent across studies.

will have considerable implications for patient care. We will seek to publish this systematic review and IPD meta-analysis in a high-impact clinical journal.

**PROSPERO registration number** CRD42018091627.

## INTRODUCTION

Mild degrees of thyroid dysfunction are common in the general population, especially among the older population.[1] Patients are usually diagnosed with subclinical thyroid dysfunction if their serum thyroid-stimulating hormone (TSH) levels are either abnormally high or low, but their serum free thyroxine (ft4)

normal.[1] A subclinical thyroid dysfunction diagnosis may be associated with several negative health outcomes; depression is one of them.

Studies that have explored the association between subclinical thyroid dysfunction and depressive symptoms have yielded conflicting results. Some observational studies report that depressive symptoms were more prevalent in patients with subclinical hypothyroidism (SHypo) than in euthyroid patients.[1 2] A cross-sectional study that included over 7500 participants found no association between SHypo and clinically relevant depression, but did find an association between subclinical hyperthyroidism (SHyper) and clinically relevant depressive symptoms.[3] Another cross-sectional study (606 participants) found that neither SHypo nor SHyper were associated with depressive symptoms.[4] The largest prospective study (220 545 participants) reported in the literature to date found no association between SHypo and incidence of depressive symptoms after 2 years of follow-up.[5] Another prospective study (606 participants) found a significant association between SHyper and depressive symptoms, but not with SHypo.[4]

These inconsistencies could be due to differences in participant's age, sex or prior depressive symptoms. By analysing individual participant data (IPD) from large cohort studies, we can define the influence of age, sex and prior depressive symptoms. IPD meta-analysis is considered to be the best way for synthesising evidence across several studies because it is not subject to ecological fallacy.[6]

## Objectives

This systematic review and IPD meta-analysis aims to determine the difference in depressive symptoms between adult participants with SHypo or SHyper and adults with a normal thyroid function (euthyroid participants) during follow-up.

## METHODS AND ANALYSIS

The protocol adheres to the preferred reporting items for systematic reviews and meta-analyses (PRISMA) statement for IPD systematic reviews[7] and to the PRISMA statement for systematic review protocols (Checklist Table in online supplementary 1.1). The protocol was registered with the International Prospective Register of Systematic Reviews (PROSPERO) on 15 May 2018.

## Eligibility criteria

We will include studies in any language and any year of publication. We will also consider unpublished studies.

### Study design

This systematic review will include prospective cohort studies that reported data on thyroid function (serum TSH and fT4 if available) at baseline, and prospective follow-up data on depressive symptoms.

We will consider any length of follow-up. Case studies or case series (ie, studies exclusively focusing on participants with thyroid dysfunction) will not be eligible. We will exclude studies that only include depressed participants,

only pregnant or postpartum participants or women who were planning to conceive. We will exclude those studies because we do not want to have depressed or pregnant women over-represented in our study. This does not mean that we will exclude single depressed or pregnant individuals from studies.

### Participants

We will include participants 18 years and older. We will exclude participants with overt hypothyroidism or hyperthyroidism from the analysis, as most indications are clear for treatment of these conditions.

### Exposed and control groups

The exposed group will be participants with SHyper or SHypo. We define SHyper as TSH level <0.45 mIU/L with normal fT4 levels and we define SHypo as TSH levels ≥4.5 mIU/L and <20 mIU/L with normal fT4 levels, as in our previous analyses.[8 9] The range for normal fT4 will be defined as study-specific reference ranges similar to our previous IPD analyses.[6 9] Participants with missing fT4 but TSH levels within the range of SHypo or SHyper will also be included in the exposed group, because most people with TSH levels in this range have subclinical rather than overt thyroid dysfunction.[10] The control group will be euthyroid participants, defined as having TSH levels between 0.45 and 4.49 mIU/L.

### Outcomes

We will consider studies with data on depressive symptoms measured on a validated continuous scale (eg, Beck Depression Inventory (BDI), Geriatric Depression Scale (GDS), etc). For the dichotomous outcome, we will consider data on incident depression among others diagnosed through International Classification of Diseases, 10th revision (ICD-10) or Diagnostic and Statistical Manual of Mental Disorders, Fifth Edition (DSM-V) codes or by established cut-off levels for incident depression on continuous depression scales.

### Study outcomes

Our primary outcome will be depressive symptoms measured on a continuous scale at the first individual available follow-up. The secondary outcomes will be depressive symptoms measured on a continuous scale at a specific follow-up time point (eg, at year 3 follow-up; decision depending on data availability) and at the last available individual follow-up. We will convert all the depressive symptoms scales to the BDI scale to have a consistent scale across the study for better interpretation of the results. The use of follow-up data as outcome adjusted for baseline instead of the change between follow-up and baseline is recommended for analysis of IPD data, especially if baseline data are balanced between groups.[11] A further secondary outcome will be the incidence of depression (binary outcome) at the first follow-up, at a specific follow-up (eg, at year 3 follow-up) and at the last available individual follow-up.

## Information sources and search strategy

We will include eligible studies from the Thyroid Studies Collaboration (TSC) and we will search the Ovid Medline, Ovid Embase, Cochrane Central Register of Controlled Trials (CENTRAL) and Cumulative Index to Nursing and Allied Health Literature (CINAHL) electronic databases from inception through May 2019. We developed the search strategy together with two experienced librarians. We developed the search strategies in Ovid Medline and translated it to match the subject headings and keywords for Ovid Embase, Cochrane CENTRAL and CINAHL. The following items were used: thyroid diseases, hyperthyroidism, hypothyroidism, thyroid hormones, triiodothyronine, thyroxine, thyrotropin, subclinical, sub-clinical, mild, subnormal, pre-clinical, preclinical and depression. The details of the search strategy are provided in online supplementary 1.2. No language restriction will be applied.

We will contact members of the TSC (www.thyroid-studies. org), an international network of cohort studies with information on thyroid status, and ask to include their unpublished data.

## Study selection

We will collect all potential articles identified through the literature search. Multiple publications from the same cohort study will be counted as one study. We will export all the identified studies to Endnote X8 and delete duplicates. First, two reviewers (LW and EM) will screen the titles and abstracts and retain articles that meet our eligibility criteria by using a Distiller Systematic Review Software. Second, two reviewers (LW and EM) will determine each remaining article's eligibility by reviewing the full text; the reviewers will keep a detailed record of reasons for excluding studies. We will resolve discrepancies by consensus with a third author (MF).

## Strategy to ask and collect individual participant data

We will use the following strategy to obtain the IPD from each cohort. We will contact the corresponding author of each study by email and ask them to participate in our study by providing IPD. If they do not respond, we will email two reminders, including the last and corresponding author. If we still do not get any answer, we will consider the collaboration as declined. If a study does not provide IPD, we will extract aggregate data from the published study and analyse it along with IPD.

We will distribute a list of all the variables that we want to include in our analysis to the responsible person from each cohort that agreed to participate. The detailed list with all the required variables is provided in online supplementary 1.3. The collaborators will use the list of variables to prepare their datasets and send them to us, or we will download the data from their online database. We will collect these datasets and merge the data into a uniform format (STATA).

## Study quality

We will assess the quality of included studies on the Newcastle-Ottawa Scale for cohort studies,[12] which allocates a maximum of nine stars to studies of the highest quality based on three parameters: selection of study groups, comparability of groups and ascertainment of the outcome of interest. Studies will be ranked either high quality (7–9 stars), moderate quality (4–6 stars) or low quality (0–3 stars).

## Data analyses

We will conduct an IPD meta-analysis in two-stages. In the first stage, we will estimate the association between depressive symptoms and participants with SHypo or SHyper and those with normal thyroid function in each cohort with multivariable regression analyses (linear regression for continuous outcomes and logistic regression for dichotomous outcomes), and adjust for depressive symptoms at baseline, age, sex, education and income. For continuous outcomes, we will use the mean difference (MD) and its relative 95% CI. For dichotomous outcomes, we will estimate the OR and its relative 95% CI.

For the continuous outcome, we will transfer all the measurements from different scales to the BDI scale by calculating a conversion factor. We will calculate the conversion factor by dividing the range of the BDI scale by the range of the scale we would like to convert. For example, to convert the Center for Epidemiological Studies Depression (CESD) scale to the BDI, we would use a conversion factor of 1.05 (63 (BDI range) ÷ 60 (CESD range)). To transfer the measurements from the CESD scale to the BDI, we will multiply each individuals CESD value by the conversion factor.

For the dichotomous outcome, we will use established cut-off values of the original continuous scales to define depression or non-depression or we use diagnosis of depression through ICD-10 or DSM-V codes among others. For example, the CESD score has the best specificity and sensitivity to define incidence ressive symptoms for a cut-off value >20.[13] In the second stage, we will pool the effect measures for each study from the first step using a random effects model.

We will assess the heterogeneity by estimating the $\tau^2$ statistic, the $I^2$ and the Q-test and explore the source of heterogeneity in several subgroups and sensitivity analyses. If we observe a high heterogeneity across study results, we will do a meta-regression including the length of follow-up for each study.

We will perform subgroup analyses by sex, age (≤57 years, >75 years), TSH levels (level 1: TSH <0.1 mIU/L, level 2: TSH 0.1–0.44 mIU/L, level 3: TSH 4.6–6.9 mIU/L, level 4: 7–9.9 mIU/L, level 5: TSH ≥10 mIU/L), ethnicity (if enough power), prior depressive episodes (yes/no) and thyroxine use at baseline (yes/no).

For the primary outcome, we will conduct several sensitivity analyses:[1] exclude participants with prior depressive episodes,[2] include studies without depressive symptoms' data at baseline,[3] exclude participants taking antidepressant medications at baseline or during follow-up,[4] exclude participants with dementia,[5] exclude participants who use thyroxine or anti-thyroid drugs at baseline and follow-up,[6] exclude participants who use thyroid-altering medication (anti-thyroid drugs, thyroxine, amiodarone, lithium),[7]

exclude participants without fT4 measured at baseline,[8] exclude studies that only provide aggregate data. As further sensitivity analysis,[9] we will calculate MDs using the original scale for each study in the first stage and pool the derived standardised MDs in the second stage.

We will assess the presence of publication bias and small study effects with funnel plots and the Egger test.[14] We will use STATA V.15 for all analyses. We will use the Grading of Recommendations, Assessment, Development and Evaluation (GRADE) approach (www.gradeworkinggroup.org) to assess the confidence on estimates.

### Patient and public involvement
Patients were not involved in the development of the research question, outcome measure and study design.

### ETHICS AND DISSEMINATION
Formal ethical approval is not required as primary data will not be collected. This study may have considerable implications for practice and help improve patient care. This systematic review and IPD meta-analysis will be submitted to a high-impact clinical journal.

#### Author affiliations
[1]Institute of Primary Health Care (BIHAM), University of Bern, Bern, Switzerland
[2]Department of General Internal Medicine, Inselspital, Bern University Hospital, University of Bern, Bern, Switzerland
[3]Department of Medicine, Division of Endocrinology, Diabetes and Metabolism, University of Pennsylvania, School of Medicine, Philadelphia, Pennsylvania, USA
[4]Department of Public Health and Nursing, K.G. Jebsen Center for Genetic Epidemiology, NTNU, Norwegian University of Science and Technology, Trondheim, Norway
[5]Department of Endocrinology, St. Olavs Hospital, Trondheim University Hospital, Trondheim, Norway
[6]Department of Internal Medicine, University of Groningen, University Medical Center Groningen, Groningen, The Netherlands
[7]Department of Clinical and Experimental Medicine, Geriatric Endocrine Unit, University Hospital of Parma, Parma, Italy
[8]Longitudinal Studies Section, Translational Gerontology Branch, Harbor Hospital, Baltimore, National Institute on Aging NIA-ASTRA Unit, Baltimore, Maryland, USA
[9]Institute for Community Medicine, Clinical-Epidemiological Research, University Medicine Greifswald, Greifswald, Germany
[10]Department of Cardiology, Leiden University Medical Center, Leiden, The Netherlands
[11]Institute of Clinical Chemistry and Laboratory Medicine, University Medicine Greifswald, Greifswald, Germany
[12]DZHK, German Centre for Cardiovascular Research, Partner Site Greifswald, University Medicine, Greifswald, Germany
[13]Section Gerontology and Geriatrics, Department of Internal Medicine, Leiden University Medical Center, Leiden, The Netherlands
[14]Department of Public Health and Center for Healthy Aging, University of Copenhagen, Copenhagen, Denmark
[15]Department of Public Health and Primary Care, Leiden University Medical Center, Leiden, The Netherlands
[16]Department of Medicine, Erasmus Medical Center, Rotterdam, The Netherlands
[17]University Hospital of Old Age Psychatry, University of Bern, Bern, Switzerland
[18]Departments of Medicine and Epidemiology & Biostatistics, University of California, San Francisco, California, USA

**Acknowledgements** The authors thank Beatrice Minder and Doris Kopp (Institute of Social and Preventive Medicine (ISPM), University of Bern, Switzerland) for helping us to develop the literature search strategy and Kali Tal, PhD (Institute of Primary Health Care (BIHAM), University of Bern, Switzerland) for editing the manuscript. This work was supported in part by the Intramural Research Program at the National Institute on Aging.

**Contributors** LW and MF will have full access to all of the data in the study and take responsibility for the integrity of the data and the accuracy of the data analysis. LW and MF will have the final responsibility for the decision to submit for publication. Concept and design: LW, SK, MF, CDG and NR. Acquisition, analysis or interpretation of data: LW, NR, CDG and MF. Drafting of the manuscript: LW, CDG, MF. Critical revision of the manuscript for important intellectual content: EM, SB, LS, ARC, BOÅ, SJLB, GC, RD, LF, HG, JWJ, MN, ST, HV, RGJW, JG, RPP, DA, DCB and NR. Statistical analysis: LW, LS and CDG. Obtained funding: NR. Administrative, technical or material support: LW, EM, SB, ARC, BOÅ, SJLB, GC, RD, LF, HG, JWJ, MN, ST, HV, RGJW, JG, RPP and DCB. Supervision: NR, CDG and MF.

**Funding** This systematic review and IPD analysis is funded by a grant from the Swiss National Science Foundation (SNSF 320030-172676 to Nicolas Rodondi). The Swiss National Science Foundation did not have any role in the design and conduct of the study; collection, management, analysis and interpretation of the data; or preparation, review or approval of the manuscript.

**Competing interests** None declared.

**Patient consent for publication** Not required.

**Ethics approval** This is a protocol paper for analysis of existing cohort data. Each study in the individual participant data set received local ethical approval. We do not plan to recruit further participants and our analysis will not include identifiable data.

**Provenance and peer review** Not commissioned; externally peer reviewed.

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
