## [Reviewer comments · BMJ Open]

ARTICLE DETAILS

TITLE (PROVISIONAL)	The Association between Subclinical Thyroid Dysfunction and Depressive Symptoms – a Protocol for a Systematic Review and Individual Participant Data Meta-Analysis of Prospective Cohort Studies
AUTHORS	Wildisen, Lea; Moutzouri, Elisavet; Beglinger, Shanthi; Syrogiannouli, Lamprini; Cappola, Anne; Asvold, Bjorn; Bakker, Stephan; Ceresini, Graziano; Dullart, Robin; Ferrucci, Luigi; Grabe, Hans; Jukema, J. Wouter; Nauck, Matthias; Trompet, Stella; Völzke, Henry; Westendorp, Rudi; Gussekloo, Jacobijn; Peeters, Robin; Klöppel, Stefan; AUJESKY, Drahomir; Bauer, Douglas; Rodondi, Nicolas; Del Giovane, Cinzia; Feller, Martin

VERSION 1 - REVIEW

REVIEWER	Huai Heng Loh Universiti Malaysia Sarawak (UNIMAS), Malaysia
REVIEW RETURNED	20-Mar-2019

GENERAL COMMENTS	This is a good study protocol. However I have a few comments: 1. By accepting participants with suppressed TSH (<0.45mIU/L as per your cut off), with missing fT4 would you be unintentionally including participants with true hyperthyroidism? 2. I would recommend you to extend your search through other search engines as well, such as OVID, PubMed, CINAHL, Cochrane etc to be more comprehensive in the search of such studies 3. What MeSH headings would you be using for the search strategy? 4. Will you include case studies or case series which fulfill your inclusion and exclusion criteria?
--

REVIEWER	zhongyan shan The first affiliated hospital of China Medical University
REVIEW RETURNED	29-Mar-2019

GENERAL COMMENTS	Based on the study, the authors present a protocol to assess the prospective association between subclinical thyroid dysfunction at baseline and depressive symptoms during follow-up by systematic review and individual participant data (IPD) analysis. In the manuscript, some problems should be noted:
--

	1. Page7 of 12, Search Strategy. The study developed the search for MEDLINE, and adapted the search for EMBASE as a supplement. The study should develop multiple databases, such as Pubmed and Cochorane in order to get sufficient materials. 2. Page5-Line34: For IPD study, why you include the aggregate data together with IPD although after sensitivity analysis given the different study design and weighting coefficient. 3. Page 8 of 12, data analyses. For the continuous outcome, the author will transfer other scales to BDI scales by calculating a conversion factor. In manuscript, the author take CESD as an example. However, some scale such as HADS(including 14 items) or PHQ-9(including 9 items) has much fewer items than BDI. There is little factor overlap between different measures, and factors are weighted differently. Was the converting factor practicable? 4. page 12 of 12, Data Request-List of Variables. The study should include data on other potential confounders such as socio-economic status and women obstetric history. 5. The study should perform subgroup for special populations, such as elderly and pregnant women.
--	--

VERSION 1 – AUTHOR RESPONSE

Reviewer(s)' Comments to Author:

Reviewer: 1

Reviewer Name: Huai Heng Loh

Institution and Country: Universiti Malaysia Sarawak (UNIMAS), Malaysia Please state any competing interests or state 'None declared': None declared

This is a good study protocol. However, I have a few comments:

Thank you for this encouraging comment.

1. By accepting participants with suppressed TSH (<0.45mIU/L as per your cut off), with missing fT4 would you be unintentionally including participants with true hyperthyroidism?

We agree that participants with overt hyperthyroidism could unintentionally be included when fT4 is missing. However, based on previous publications, we expect that only 12% of those who have a TSH<0.45mIU/L do have an elevated fT4 (i.e. overt hyperthyroidism) (1). Nevertheless, we will conduct a sensitivity analysis excluding those with missing fT4.

(page 6, lines 289-290)

”(7) exclude participants without fT4 measured at baseline.”

2. I would recommend you to extend your search through other search engines as well, such as OVID, PubMed, CINAHL, Cochrane etc to be more comprehensive in the search of such studies

Thanks for pointing this out. We additionally included the CINAHL and Cochrane database into the protocol.

(page 5, lines 194-195)

“We will include eligible studies from the Thyroid Studies Collaboration and we will search the Ovid Medline, Ovid Embase, Cochrane CENTRAL and CINAHL electronic databases from inception through April 2019.”

We decided to perform the search for Medline via OVID instead of Pubmed. We actually used OVID to access both the Medline and Embase database. In order to make that clear we mentioned that we searched in Ovid Medline and Ovid Embase

(page 5, lines 207-209)

“We developed the search strategies in Ovid Medline and translated it to match the subject headings and keywords for Ovid Embase, Cochrane CENTRAL and CINAHL.”

3. What MeSH headings would you be using for the search strategy?

We provide the search strategies including the MeSH terms in the supplementary of the protocol. In addition, we added the search strategy including the MeSH terms to the Methods of the paper (page 5, lines 209-211)

“The following items were used: thyroid diseases, hyperthyroidism, hypothyroidism, thyroid hormones, triiodothyronine, thyroxine, Thyrotropin, subclinical, sub-clinical, mild, subnormal, preclinical, preclinical, depression. The details of the search strategy are provided in Supplementary 1.2.”

4. Will you include case studies or case series which fulfill your inclusion and exclusion criteria?

Thank you for raising this point. Case studies or case series (i.e. studies exclusively focusing on participants with thyroid dysfunction) are not eligible as our goal is to compare participants with thyroid dysfunction to euthyroid participants within each cohort.

To make it more clearly, we added the following sentence

(page 4, lines 166-167)

“Case studies or case series (i.e. studies exclusively focusing on participants with thyroid dysfunction) will not be eligible.”

Reviewer: 2

Reviewer Name: Zhongyan Shan

Institution and Country: The first affiliated hospital of China Medical University Please state any competing interests or state 'None declared': None declared

1. Page7 of 12, Search Strategy. The study developed the search for MEDLINE, and adapted the search for EMBASE as a supplement. The study should develop multiple databases, such as Pubmed and Cochorane in order to get sufficient materials.

Thanks for highlighting this. We agree, and we refer to our answer to point 2 of reviewer 1 above.

2. Page5-Line34: For IPD study, why you include the aggregate data together with IPD although after sensitivity analysis given the different study design and weighting coefficient.

Thanks for giving us the opportunity to clarify. Our intention is to make full use of all available evidence. Therefore, we also include in our main analysis cohort studies only providing aggregate data (i.e. no IPD). Then, in a sensitivity analysis, we will exclude cohort studies that do not provide IPD, thereby testing the robustness of the results of our main analysis. We included the following sensitivity analysis.

(page 6, line 290)

“(8) exclude studies that only provide aggregate data.”

3. Page 8 of 12, data analyses. For the continuous outcome, the author will transfer other scales to BDI scales by calculating a conversion factor. In manuscript, the author take CESD as an example. However, some scale such as HADS(including 14 items) or PHQ-9(including 9 items) has much fewer items than BDI. There is little factor overlap between different measures, and factors are weighted differently. Was the converting factor practicable?

We agree that transforming other scales to BDI is a limitation. Therefore, we mention this point in the limitation section. The reason why we chose to express results on the BDI scale is that clinical readers will be able to easily interpret the results of this IPD analysis since the BDI is one of the most widely used scales in clinical practice (2). In addition, we will perform a sensitivity analysis in which we will use standardised mean differences to express results instead of BDI.

(page 7, lines 303-304)

“(9) we will calculate mean differences using the original scale for each study in the first stage and pool the derived standardized mean differences in the second stage. “

For the secondary outcome (i.e. incidence of depression), we do not transform other scales to BDI, but use the original scale from each study, and apply established, scale-specific cut-off values to ascertain depression.

4. page 12 of 12, Data Request-List of Variables. The study should include data on other potential confounders such as socio-economic status and women obstetric history.

Thanks for this suggestion. We will now add income and education to our adjusted models. We wrote (page 6, line 264)

“...and adjust for depressive symptoms at baseline, age, sex, education, and income”.

We will not adjust for obstetric history for the following reasons: It is not an established risk factor for depression, and it is not associated with thyroid dysfunction. Yet, we acknowledge that the selection of confounder variables is notoriously difficult, and – to some extent – arbitrary.

5. The study should perform subgroup for special populations, such as elderly and pregnant women.

We confirm that we plan subgroup analyses according to age (among others). We wrote (page 6, lines 282-284)

“We will perform subgroup analyses by sex, age (≤ 57 years, >75 years) , TSH levels (Level1: TSH <0.1 mIU/L, Level 2: TSH 0.1-0.44 mIU/L, Level 3: TSH 4.6-6.9 mIU/L, Level 4: 7-9.9 mIU/L, Level 5: TSH ≥ 10 mIU/L), ethnicity (if enough power), prior depressive episodes (yes/no) and thyroxine use at baseline (yes/no).”

Based on a previous IPD with the same/similar cohorts (3), we know that the median age of participants from these cohorts ranges from 51 to 85 years. Therefore, we expect the number of pregnant women to be very low, precluding a meaningful subgroup analysis.

References:

1. Schneider C, Feller M, Bauer DC, Collet TH, da Costa BR, Auer R, et al. Initial evaluation of thyroid dysfunction - Are simultaneous TSH and ft4 tests necessary? PLoS One.

2018;13(4):e0196631.

2. Hawley CJ, Gale TM, Smith PS, Jain S, Farag A, Kondan R, et al. Equations for converting scores between depression scales (MADRS, SRS, PHQ-9 and BDI-II): good statistical, but weak idiographic, validity. Hum Psychopharmacol. 2013;28(6):544-51.

3. Blum MR, Bauer DC, Collet TH, Fink HA, Cappola AR, da Costa BR, et al. Subclinical thyroid dysfunction and fracture risk: a meta-analysis. JAMA. 2015;313(20):2055-65.

VERSION 2 – REVIEW

REVIEWER	Huai Heng Loh University Malaysia Sarawak, Malaysia
REVIEW RETURNED	13-Jun-2019

GENERAL COMMENTS	Revision is adequate. Good work.
----------------------------------

REVIEWER	zhongyan shan The first affiliated hospital of china medical university,china
REVIEW RETURNED	05-Jun-2019

GENERAL COMMENTS	The authors have made corresponding modifications or explanations according to the opinions of reviewers. The parts that cannot be modified have been included in limitations.
--